# Transient Expression of Glycosylated SARS-CoV-2 Antigens in *Nicotiana benthamiana*

**DOI:** 10.3390/plants11081093

**Published:** 2022-04-18

**Authors:** Valentina Ruocco, Richard Strasser

**Affiliations:** Department of Applied Genetics and Cell Biology, Institute of Plant Biotechnology and Cell Biology, University of Natural Resources and Life Sciences, Muthgasse 18, A-1190 Vienna, Austria; valentina.ruocco@boku.ac.at

**Keywords:** COVID-19, glycoprotein, glycosylation, molecular farming, plant expression system, virus

## Abstract

The current COVID-19 pandemic very dramatically shows that the world lacks preparedness for novel viral diseases. In addition to newly emerging viruses, many known pathogenic viruses such as influenza are constantly evolving, leading to frequent outbreaks with severe diseases and deaths. Hence, infectious viruses are a recurrent burden to our daily life, and powerful strategies to stop the spread of human pathogens and disease progression are of utmost importance. Transient plant-based protein expression is a technology that allows fast and highly flexible manufacturing of recombinant viral proteins and, thus, can contribute to infectious disease detection and prevention. This review highlights recent progress in the transient production of viral glycoproteins in *N. benthamiana* with a focus on SARS-CoV-2-derived viral antigens.

## 1. Introduction

The increasing emergence of infectious diseases caused by viruses constitutes a significant threat to human and animal health. The fast and global spread of Severe Acute Respiratory Syndrome Coronavirus-2 (SARS-CoV-2), the causative pathogen of COVID-19, resulted in an unprecedented worldwide crisis with million deaths [1]. In the last two decades, three highly pathogenic coronaviruses, SARS-CoV-1, SARS-CoV-2, and Middle Eastern Respiratory Syndrome (MERS)-CoV eventually crossed the species barrier to infect humans and cause pathogenesis [2,3]. Spurred on by more frequent contact with wildlife animals, urbanization, and climate change, it appears inevitable that similar zoonotic or vector-borne transmission events will occur in the near future and cause comparable epidemics and pandemics. Newly arising SARS-CoV-2 variants such as Omicron, which result in immune escape and faster transmission of the virus [4], demonstrate that our countermeasures to fight the virus require constant and rapid adaptations. In this light, new diagnostic, therapeutic, and vaccination approaches must be quickly available to keep pace with evolving viruses and prevent devastating waves of infections. Transient plant-based protein expression allows rapid and highly flexible manufacturing of recombinant viral proteins and, thus, contributes to solving current and emerging global health challenges [5]. Plant-based recombinant protein production is cost-efficient, safe, easily scalable and, thus, very attractive for developing countries with limited infrastructure [6,7]. The use of plants or plant cells for production of monoclonal antibodies, vaccines and other recombinant proteins is well established, and many studies have proven the functionality of the plant-produced material in animals or clinical trials [8,9,10,11]. 

Most enveloped human and veterinary viruses have a glycoprotein coat consisting of envelope or spike proteins that are embedded in a lipid layer derived from the infected host cell [12]. Glycosylation of these viral proteins plays an important role in their folding, virus entry, and the host immune response to infection [12,13]. In addition to the manufacturing of antigens as diagnostic reagents or for vaccine development [14,15], recombinant viral glycoproteins with defined glycans are important tools for research, for example, to guide design for mRNA vaccine technologies [16,17]. Here, we highlight recent advancements and limitations in the transient production of heterologous viral glycoproteins in *Nicotiana benthamiana*, a tobacco-related species that is widely used for plant-based recombinant protein expression.

## 2. Transient Expression

The production of heterologous proteins in plants can be achieved via stable or transient expression. The production of transgenic plants for protein expression is time-consuming, protein yields are often low, and adaptation of the expression system to changes in the protein of interest, for example, to produce a slightly different recombinant protein, requires a repetition of the whole process [18]. This makes transgenic plants as production hosts for viral antigens of emerging or evolving viruses impracticable. By contrast, transient expression is fast and flexible, which allows high-throughput screening of numerous expression constructs and simple scalability from research-scale (e.g., 100–500 g biomass) to large-scale manufacturing (e.g., >500 kg biomass). Due to these advantages, efficacious recombinant viral antigens or virus-like particles can be produced at high levels within a few weeks [19]. 

*Agrobacteria* are transformed with expression vectors for recombinant proteins, and transient expression is achieved by *Agrobacterium*-mediated infiltration (agroinfiltration) of *N. benthamiana* leaves using a syringe (research scale) or vacuum. *Agrobacteria* carry DNA that is delivered to the nucleus of plant cells for the production of mRNA and the protein of interest. Recombinant proteins are purified a few days after agroinfiltration from infiltrated leaves (Figure 1). 

*N. benthamiana* is the preferred host for transient protein production because this nonfood plant is highly susceptible to virus infection [20], and leaves can easily be infiltrated. Furthermore, *N. benthamiana* has rather simple growth requirements, produces high biomass, and is amenable to modifications by gene silencing or genome editing. Transgenic expression of two RNAi hairpin constructs resulted, for example, in the generation of glycoengineered *N. benthamiana* ΔXT/FT plants that have silenced expression of β1,2-xylosyltransferases and core α1,3-fucosyltransferases genes [21]. This plant line is used worldwide by academia and industry to produce recombinant glycoproteins with complex *N*-glycans that are almost completely devoid of core α1,3-fucose and β1,2-xylose residues and resemble human-type complex *N*-glycans. Notably, the ZMapp monoclonal antibody cocktail that was used for treatment of Ebola virus infections was produced in ΔXT/FT plants [22,23]. Genome editing has recently been used to generate *N. benthamiana* that are completely devoid of functional β1,2-xylosyltransferases and core α1,3-fucosyltransferases [24]. Under greenhouse conditions, the knockout plants grew normally and did not display any phenotype which is consistent with previous studies, showing that *N. benthamiana* tolerates stable glycoengineering very well [21,25,26]. The detailed annotation of the *N. benthamiana* genome, transcriptome, and proteome [27,28] makes it more straightforward to overcome current bottlenecks in the expression host and perform multiple genome-editing steps to inactivate whole gene families that contribute to unwanted posttranslational modifications or proteolytic processing.

In addition to achievements in glycoengineering and genome-editing technologies, a number of powerful expression vectors for transient expression in *N. benthamiana* have been developed in the last 20 years. Commonly used expression vectors contain elements from plant viruses for high-level expression in leaves. Among the most widely used vectors are those carrying cauliflower mosaic virus (CaMV) regulatory sequences (e.g., pTRA [29]), cowpea mosaic virus-derived (CPMV) vectors (e.g., pEAQ-*HT* [30]), potato virus X (PVX)- and tobacco mosaic virus (TMV)-based vectors (e.g., magnICON [31]), or bean yellow dwarf geminivirus-derived DNA replicon vectors [32,33].

## 3. Protein Glycosylation

Glycosylation is a fundamental co- or posttranslational modification of proteins. The attachment of a preassembled oligosaccharide or an individual sugar onto proteins can have a major impact on many cellular processes and very often diversifies the functions of the modified proteins. Common modifications such as *N*- and *O*-glycosylation exert a major influence on the immunogenicity of a viral protein by affecting the protein conformation or by shielding of epitopes at the surface of the protein. Glycosylation and processing of glycans depends on structural properties of the viral glycoprotein and host cell factors including the glycosylation machinery of the expressing cell and the trafficking route through the secretory pathway. Since exposed envelope or spike glycoproteins are principal targets of the host immune response [34,35], it is crucial to use protein production systems for viral glycoproteins that allow the control of glycosylation and the production of authentic glycan structures. Glycosylation should, therefore, be considered as an important parameter in antigen design for subunit vaccine development or for components in serological diagnostic tests. Recent advances in glycoengineering in plants enable the production of custom-made glycoproteins carrying homogeneous human-type *N*-glycans and the controlled modulation of glycosylation for structure–function studies [36]. Notably, plants have a much simpler *N*-glycan processing pathway which can be altered in *N. benthamiana* without affecting the overall plant development, biomass production, or recombinant protein production. Furthermore, since plants lack a mammalian mucin-type *O*-glycan biosynthesis pathway, they offer another advantage over mammalian cells and can be engineered to facilitate the stepwise assembly of defined *O*-glycans by expression of the required glycosyltransferases without any interference from mammalian glycosylation pathways [37,38,39].

During viral infections, viruses enter cells by binding to their cognate cell surface receptors. For instance, SARS-CoV-2 cell entry is dependent on the heavily glycosylated trimeric spike protein via recognition of the host cell receptor angiotensin-converting enzyme 2 (ACE2). The spike protein is a type I transmembrane protein with a large N-terminal ectodomain that protrudes from the viral surface, and the monomeric spike protein harbors 22 *N*-glycans and several potential *O*-glycosylation sites [40,41,42]. For coronaviruses, *N*-glycans are important for spike protein folding, for modulating the accessibility to host proteases, for shielding to avoid the detection by the immune response of infected individuals, and for interaction with cellular receptors [43,44,45,46]. The loss of the conserved *N*-glycosylation site at asparagine 370 in the SARS-CoV-2 spike protein affects the conformation of the receptor-binding domain (RBD) resulting in a higher affinity to ACE2 and enhanced virus replication [47,48]. A reacquisition of this *N*-glycan by mutation provides a means to evade an established immune response [49]. Such an event may occur in the future, and the *N*-glycan at N370 may then act as a barrier against neutralizing antibodies. A role of *N*-glycosylation in immune evasion by camouflaging immunogenic protein epitopes is well known for pathogenic viruses [35,50], and the presence or absence of *N*-glycans can have positive or detrimental effects on the virus infectivity and antigenicity [51,52]. These selected examples demonstrate the importance of glycosylation for viruses and rational antigen design.

## 4. Pathways for *N*- and O-Glycan Processing

The initial step of *N*-glycosylation is the *en bloc* transfer of a preassembled oligosaccharide (Glc_3_Man_9_GlcNAc_2_) from a lipid-linked carrier to an asparagine residue present in a consensus sequence (Asn–X–Ser/Thr, X can be any amino acid except proline) on nascent polypeptides [53]. The oligosaccharyltransferase (OST) complex catalyzes the transfer of the assembled oligosaccharide to the nascent polypeptide in the lumen of the endoplasmic reticulum (ER) (Figure 2). Immediately after the transfer, two glucose residues are sequentially removed from the oligosaccharide by α-glucosidase I (GCSI) and II (GCSII). The remaining glucose residue on the oligomannosidic *N*-glycan allows the viral glycoproteins to associate with the lectin chaperones calnexin (CNX) and calreticulin (CRT). Together with other proteins like protein disulfide isomerases, CNX/CRT promote the folding of glycoproteins. Iminosugar inhibitors that block α-glucosidases and interfere with CNX/CRT binding have, therefore, been investigated as drugs to prevent native viral glycoprotein folding and subsequent virus assembly and propagation [53,54]. 

Following trimming of the remaining glucose residue by GCSII, glycoproteins that have acquired their native conformation are released from the CNX/CRT quality control cycle and allowed to exit the ER to the Golgi apparatus. In the next *N*-glycan processing step, a single mannose residue is removed from the middle branch of the oligomannosidic *N*-glycan by the ER-α-mannosidase I (MNS3) to form the Man_8_GlcNAc_2_ *N*-glycan [55,56]. Golgi α-mannosidase I (GMI) cleaves off three additional mannose residues and generates the acceptor substrate for *N*-acetylglucosaminyltransferase I (GNTI) which adds a GlcNAc to the core α1,3-mannose residue. GNTI has a central function in *N*-glycan processing, as the transferred GlcNAc residue is a prerequisite for further processing and complex *N*-glycan formation [57]. These early steps of *N*-glycan processing that take place in the ER and *cis*/medial Golgi are conserved between mammals and plants. 

After the action of GNTI, the plant *N*-glycans are typically trimmed by Golgi α-mannosidase II (GMII), and *N*-acetylglucosaminyltransferase II (GNTII) transfers a GlcNAc to the α1,6-mannose residue. In addition, β1,2-xylosyltransferase (XYLT) and core α1,3-fucosyltransferase (FUT) use the processed structures as substrates to generate typical plant complex *N*-glycans (GlcNAc_2_XylFucMan_3_GlcNAc_2_ also termed GnGnXF) [58]. Further elongation is only found on a rather small number of plant proteins, and these so-called Lewis A structures (Fucα1,4-(Galβ1,3-GlcNAc-R) are generated in the *trans* Golgi by β1,3-galactosyltransferase 1 (GALT1) and α1,4-fucosyltransferase (FUT13) [59]. 

In mammalian cells, the innermost GlcNAc residue of complex *N*-glycans is frequently modified by core α1,6-fucosyltransferase (FUT8) which transfers a fucose in α1,6-linkake instead of the α1,3-fucose that is attached by plant FUT. *N*-Acetylglucosaminyltransferases IV and V initiate branching of the *N*-glycan structures, and *N*-acetylglucosaminyltransferase III can add a bisecting GlcNAc to complex *N*-glycans. While this bisecting GlcNAc is not further elongated, the other GlcNAc residues can be modified with galactose mainly in β1,4-linkage and terminal sialic acid residues (Figure 2). All these complex *N*-glycan modifications are not present in plants, but *N. benthamiana* can be engineered to produce mammalian-type complex *N*-glycan structures including core α1,6-fucosylation, the attachment of β1,4-linked galactose, branching of *N*-glycans, and the incorporation of sialic acid in different linkages [36,60].

While the *N*-glycan processing machinery is partly conserved between plants and mammals, *O*-glycosylation is fundamentally different [61]. In mammals, the most common *O*-glycosylation found on secretory proteins is the posttranslational incorporation of a single *N*-acetylgalactosamine (GalNAc) to specific serine or threonine residues (Figure 3). This reaction is catalyzed by a large family of polypeptide *N*-acetylgalactosaminyltransferases (GalNAc-T) [62]. The single GalNAc can be further modified by the stepwise attachment of different monosaccharides including galactose, GlcNAc, fucose, and sialic acids. These modifications result in a range of structurally diverse mucin-type core *O*-glycans. Polypeptide *N*-acetylgalactosaminyltransferases and other glycosyltransferases involved in mucin-type *O*-glycan elongation in mammals are absent from plants. Plants, on the other hand, can modify serine residues with a single galactose, and prolines can be hydroxylated and subsequently modified to carry short arabinose chains or larger arabinogalactan glycan moieties [61]. Transient expression of the mammalian mucin-type *O*-glycosylation pathway in *N. benthamiana* resulted in the formation of different human-type *O*-glycan core structures on recombinant proteins such as IgA1 or EPO-Fc [37,63]. Virus envelope and spike proteins can be highly *O*-glycosylated with very heterogeneous structures that also contribute to epitope shielding and interaction with host cells [64]. As observed for other recombinant human glycoproteins [38,65], it is likely that exposed proline residues of transiently expressed viral glycoprotein will be converted to hydroxyproline and further modified with arabinose residues. For subunit vaccine development or therapeutic applications of recombinant proteins, these nonhuman modifications are unwanted and could lead to adverse immune reactions [66]. However, this limitation of the plant-based production platform may be overcome by identification and elimination of the corresponding prolyl-4-hydroxylases (P4Hs) that catalyze hydroxyproline formation on recombinant glycoproteins [67].

The impact of protein glycosylation on viral glycoprotein function is well known. For many viruses, different glycan-mediated interactions play a role at the host–pathogen interface. The envelope proteins of HIV-1, Ebola, and herpes simplex virus are modified by *O*-linked glycosylation, which can affect the binding to cellular receptors [68] and directly or indirectly influence various steps of viral infection and replication. Together with *N*-glycans, distinct *O*-glycans potentially present ligands for innate immune receptors that bind in a glycan-dependent manner and subsequently promote the interaction with the primary cellular receptors. For example, C-type lectin receptors such as DC-SIGN present in epithelial and endothelial cells interact with the oligomannosidic and complex *N*-glycans on the SARS-CoV-2 spike protein [69,70]. Thus, for many viruses, *N*- and *O*-glycans contribute to virus infection and play a role at different steps of the viral life cycle and for the host immune response.

## 5. Transiently Expressed Glycosylated SARS-CoV-2 Antigens

Soon after the SARS-CoV-2 outbreak was declared a pandemic, several studies showed that transient production of SARS-CoV-2 antigens is possible in *N. benthamiana* within a short time [14,71,72]. Recombinant RBD of the SARS-CoV-2 spike protein is frequently used as antigen in serological assays [73] and for vaccine development because RBD is the primary target for potent virus-neutralizing antibodies [15,74]. Recombinant plant-produced and human HEK293 cell-produced RBD displayed comparable binding affinities to ACE2 [75,76]. Furthermore, plant-derived RBD was recognized by a conformation-dependent RBD-specific antibody [71,77] and by sera from SARS-CoV-2-infected individuals [14,75,78]. Taken together, these studies very impressively demonstrate the functionality of the plant-produced SARS-CoV-2 antigen. 

The SARS-CoV-2 RBD (most widely used sequence domain: R319–F541) has two *N*-glycosylation sites (N331 and N343) that are glycosylated with complex *N*-glycans when expressed in *N. benthamiana* [77,79]. The two RBD *N*-glycans are not part of the receptor-binding motif and not directly involved in receptor binding, which is consistent with comparable ACE2 binding affinities for glycoengineered RBD produced in the ΔXT/FT line, in wildtype *N. benthamiana* or in HEK293 cells. However, the two SARS-CoV-2 RBD *N*-glycans are crucial for protein folding and efficient production in plants [77]. A slightly truncated RBD protein (amino acids 319–533) that lacks an unpaired cysteine at position C538, could not be produced as soluble protein when both or individual N-glycosylation sites were mutated, suggesting that both *N*-glycans contribute to proper folding of RBD in plants [77].

Comparison of the seroreactivity to the RBD carrying *N*-glycans with β1,2-xylose and core α1,3-fucose (GnGnXF structures) and to the RBD lacking these sugar residues (GlcNAc_2_Man_3_GlcNAc_2_, GnGn structures) showed that both RBD glycoforms are suitable for serological assays. A minor limitation for production of glycosylated viral antigens in wildtype plants is a potential higher number of false-positive results with sera containing allergy-related cross-reactive carbohydrate determinant (CCD) antibodies [75]. In addition to functional testing in serological assays, the immunogenicity of *N. benthamiana*-produced RBD variants has been examined by immunization of mice [76,80,81] and cynomolgus monkeys [80]. Immunization with plant-produced RBD protein resulted in the generation of SARS-CoV-2 antibodies that could neutralize the virus in cell-based assays. Mamedov et al. (2021) showed that a deglycosylated RBD variant could elicit high titers of antibodies with a potent virus-neutralizing activity similar to the glycosylated RBD [76]. The deglycosylation of RBD was achieved by co-expression of an ER-targeted endoglycosidase (Endo H) which cleaves between the chitobiose core of oligomannosidic *N*-glycans. Thus, the protein is not fully deglycosylated, and a single GlcNAc is still present at the *N*-glycosylation site. Notably, the Endo H modified RBD displayed almost similar affinity to ACE2 as the fully glycosylated RBD carrying oligomannosidic *N*-glycans [76]. The findings are consistent with the observation that RBD *N*-glycans are essential for proper folding in the ER [77], but not for ACE2 binding and the presentation of immunogenic epitopes [15,75,76,82,83]. The SARS-CoV-2 nucleoprotein (N protein) was also expressed in the ER in *N. benthamiana*. The viral nucleoprotein is involved in packaging of the viral RNA in the cytosol and is normally not in contact with the glycosylation machinery of the host. Despite having five potential *N*-glycosylation sites, the nucleoprotein targeted to the secretory pathway appeared not glycosylated [84]. Whether the recombinant nucleoprotein was correctly folded in the milieu of the ER was not further investigated, but an antigen cocktail comprising RBD and the nucleoprotein elicited high-titer antibodies in mice, suggesting that immunogenic epitopes were preserved. A truncated SARS-CoV-2 nucleoprotein comprising the C-terminal region with a single *N*-glycosylation site was expressed in the secretory pathway [85]. The produced protein was functional in serological assays and showed high diagnostic sensitivity and specificity for SARS-CoV-2 antibody detection.

For RBD expression, slightly different regions from the S1 domain of the spike protein were expressed in plants, and different purification tags (e.g., Fc, His, FLAG tags) and purification approaches (crude protein extracts or apoplastic fluid) were used. Furthermore, in some studies, the KDEL Golgi-to-ER retrieval signal was attached to the C-terminus for retention in the ER [72,76,77]. However, in all these studies on transient RBD production in *N. benthamiana*, similar expression levels (up to 100 µg/g fresh weight) and yields after purification (ranging from 2 to 25 µg/g fresh weight) were reported. While viral antigens often give low yields, expression levels of monoclonal IgG antibodies can be in the range of 1 mg/g or higher [86,87]. Therefore, economically feasible manufacturing of recombinant SARS-CoV-2 antigens may require further optimization and engineering of the secretory pathway [88]. With one exception, all current studies related to RBD expression focused on the original sequence from the SARS-CoV-2 Wuhan isolate. RBD-Fc variants corresponding to the alpha and beta SARS-CoV-2 variants have been expressed in *N. benthamiana* [89]. While both variants displayed binding to ACE2, the plant-produced beta variant displayed differences in the reactivity with neutralizing antibodies. This finding likely reflects sequence intrinsic effects on the conformation of epitopes in the beta variant rather than conformational changes caused by the folding machinery of the expression host.

Although most studies have focused on the expression of spike protein domains, the SARS-CoV-2 genome encodes other *N*-glycosylated proteins, such as the envelope (E) and membrane (M) proteins [90]. The functions of their *N*-glycans for SARS-CoV-2 have not yet been studied in detail. In *N. benthamiana*, co-expression of native forms of the SARS-CoV-2 spike protein with the envelope and membrane proteins resulted in the formation of VLPs that were recognized by antibodies in sera from convalescent patients [91]. Such noninfectious plant-made VLPs were also obtained by co-expressing the envelope, membrane, and nucleocapsid proteins [92]. In both studies, the *N*-glycosylation status of the viral glycoproteins was not assessed.

The biopharmaceutical company Medicago designed a VLP (CoVLP) using the SARS-CoV-2 spike protein as an antigen. This VLP-based vaccine was produced using a manufacturing platform that was developed for a quadrivalent VLP influence vaccine [93]. The VLP mimics the surface structure of the natural virus with an antigenic moiety but is noninfectious and nonreplicating as it lacks the replicating viral RNA. Hence, VLPs are generally considered safe and can stimulate an immune response against the virus when administered to humans. In the CoVLP, the SARS-CoV-2 spike protein is presented in the prefusion conformation that closely resembles the native structure in the virus [94]. A few mutations have been introduced to increase the stability of the spike protein, but none of these changes directly affects an *N*-glycosylation site. Hence, a spike monomer in the CoVLP vaccine carries all 22 *N*-glycosylation sites. This VLP is, therefore, heavily glycosylated and supposed to harbor numerous complex *N*-glycans carrying β1,2-xylose and core α1,3-fucose when produced in *N. benthamiana* wildtype plants. Importantly, the reported positive outcome from the clinical studies shows that the plant-produced vaccine is safe without any severe allergic reactions and efficacious against SARS-CoV-2 including the Delta variant of concern [94,95,96,97]. On 24 February 2022, Medicago in collaboration with GlaxoSmithKline (GSK) announced that Health Canada has granted approval for COVIFENZ^®^, a COVID-19 vaccine based on *N. benthamiana*-produced CoVLP.

## 6. Glycan Processing, *N*-Glycosylation Site Occupancy, and Strategies to Overcome Current Limitations of Viral Glycoprotein Expression in Plants

Given the importance of glycosylation for virus infection and the immune response of the host, it is crucial to overcome shortcomings of the plant-based production system for viral glycoproteins. A comparative analysis of the site-specific *N*-glycosylation of recombinant viral glycoproteins expressed in *N. benthamiana* and mammalian cells revealed reduced *N*-glycan processing and under-glycosylation at certain *N*-glycosylation sites [98] (Figure 4). In addition, many heavily glycosylated viral antigens such as the ectodomain of the SARS-CoV-2 spike [99] or HIV envelope proteins are expressed in plants at low levels [100].

For *N*-glycans, the degree of glycan processing (e.g., oligomannosidic vs. specific complex-type modifications) is influenced by the architecture of the viral glycoprotein and the resulting accessibility of glycans to the processing glycosidases and glycosyltransferases. Viral glycoproteins often display a large number of oligomannosidic *N*-glycans that are not further processed to complex *N*-glycans [41]. The reduced processing of these *N*-glycans can be caused by the presence of dense patches of *N*-glycans with glycan–glycan contacts that abolish access by α-mannosidases and initiation of complex *N*-glycan formation by GNTI [101]. For SARS-CoV-2, a comparison of *N*-glycans from mammalian cell-derived RBD, monomeric, trimeric, or virion-derived spike revealed an overall conservation of *N*-glycan occupancy and *N*-glycan processing, with some minor differences in *N*-glycan composition on distinct sites [102,103]. Human HEK293 cell-derived monomeric RBD displayed complex *N*-glycans with core α1,6-fucose and sialylation on both glycosylation sites. By contrast, on spike derived from purified virus, core fucosylation was reduced, and the majority of the *N*-glycans lacked sialic acid and were galactosylated. Recombinant SARS-CoV-2 RBD produced in wildtype *N. benthamiana* showed typical plant complex *N*-glycans (GnGnXF), and, in glycoengineered ΔXT/FT *N. benthamiana*, mainly complex-type GnGn *N*-glycans were detected on both *N*-glycosylation sites [76]. This provides evidence that the site-specific processing of the *N*-glycans on the RBD of the SARS-CoV-2 spike is conserved when produced in plants. The predominant GnGn structure is optimally suited to engineer variants with defined complex *N*-glycans including Lewis A or blood group antigen-containing structures [75,79]. A site-specific analysis of recombinant SARS-CoV-2 S1, full-length monomeric or trimeric spike expressed in plants has not been reported yet. However, according to the data for the conserved accessibility of distinct N-glycans when expressed in different animal cells, we anticipate that under-processed glycans (e.g., the oligomannosidic *N*-glycan at site N234) in the virus-derived spike will also be under-processed when expressed in plants. The transient co-expression of glycosyltransferases for core α1,6-fucosylation, β1,4-galactosylation, and attachment of sialic acid will enable the production of viral glycoproteins with desired *N*-glycan structures for functional characterization or specific applications in plants. Furthermore, glycoengineering can be used to produce viral antigens with glycan epitopes that are recognized by naturally circulating antibodies [79]. For example, distinct blood group antigens present on viral glycoproteins may be recognized by naturally occurring anti-glycan antibodies, and the antibody binding may block the virus transmission and infection of cells [104,105]. In future applications, immunization with viral antigens carrying distinct glycan epitopes may be used to elicit an immunostimulatory effect [106,107]. Glycoengineering combined with transient expression in *N. benthamiana* is very attractive for applications aiming to optimize the safety, functionality, and efficacy of protein-based vaccines.

While the function of the OST complex is conserved in all eukaryotes, there are yet unknown differences in plants that cause under-glycosylation of transiently expressed recombinant proteins. For immunoglobulins such as IgG, 10–20% of the single *N*-glycosylation site in the heavy chain is often non-glycosylated [108]. For other glycoproteins, including heavily glycosylated HIV antigens, some *N*-glycosylation sites are almost completely unoccupied and display aberrant glycosylation [98]. A high proportion of under-glycosylation has two important consequences for viral glycoproteins. Firstly, the lack of an *N*-glycan at a distinct position leads to accessibility of the polypeptide surface that is otherwise shielded by the glycan, which potentially results in an off-target immune response [109]. Secondly, since *N*-glycans are directly or indirectly involved in protein folding, under-glycosylation at distinct sites affects proper folding of the recombinant protein and, thus, the expression levels and protein function. The direct effects are potentially related to reduced solubility, conformational changes, and an increased tendency to aggregate. The indirect effect is likely caused by an impaired interaction with the lectin chaperones CNX/CRT. The lack of an *N*-glycan from SARS-CoV-2 has a huge impact on RBD expression in *N. benthamiana* [77]. Notably, human CRT overexpression rescued the expression of RBD carrying a single *N*-glycan, which shows the dependency of the plant-produced viral protein on glycan-mediated protein folding. The beneficial effect of human CRT has also been demonstrated for other viral glycoproteins expressed in *N. benthamiana*, showing that lectin chaperone expression is a general approach to improve protein yields and reduce constraints that arise from the expression of heavily glycosylated heterologous proteins [99,100]. In future studies, it will be interesting to examine how the CNX/CRT co-expression affects *N*-glycan processing, recombinant protein secretion, and the overall functionality of the expressed viral antigens. As shown for antibodies, under-glycosylation of viral glycoproteins may be overcome by transient co-expression of a single-subunit OST from *Leishmania major* (LmSTT3D) [108]. Further studies are needed to combine different approaches and increase the quality and quantity of viral glycoproteins when transiently expressed in *N. benthamiana*.

## 7. Conclusions

The recent examples of SARS-CoV-2 viral antigen expression in *N. benthamiana* demonstrate that plant-produced viral antigens can be used in serological assays to monitor the immune surveillance, or as VLPs and protein subunit vaccines to elicit specific immune responses. A number of plant-produced SARS-CoV-2 antigens and VLPs are in preclinical and clinical trials [11]. The spike-containing VLP-based vaccine from Medicago has shown efficacy and safety in phase 3 clinical trials [97] and has been approved, which is a great success and a huge breakthrough for plant-based manufacturing of vaccines and therapeutic proteins. Compared to mammalian cell-based expression systems, the transient expression in *N. benthamiana* provides a safe, simplified, and scalable manufacturing approach. Investment in infrastructure in low- and middle-income countries, and build-up of regional plant-based manufacturing capacity should help to increase the global access to reagents and therapeutics, reduce the timelines for production, and potentially result in more affordable products with the aim of increasing global readiness for the next pandemic.

## Figures and Tables

**Figure 1 plants-11-01093-f001:**
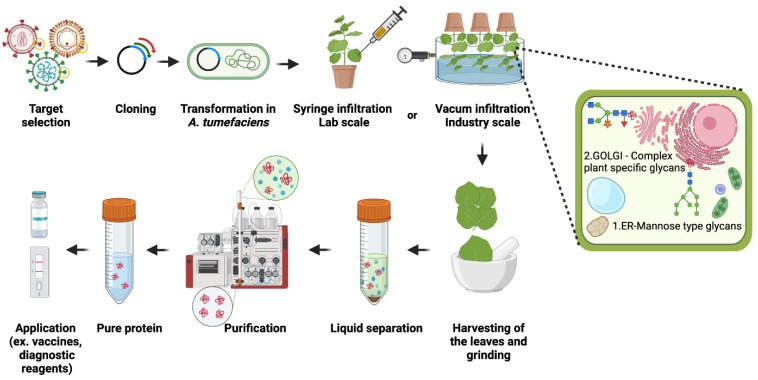
Schematic representation of the approach to transiently produce viral antigens in *N. benthamiana*. Upon selection of the appropriate target viral protein (e.g., the spike protein from SARS-CoV-2), codon-optimized synthetic DNA fragments coding for the viral protein are cloned into an expression vector, *Agrobacteria* are transformed, and the bacterial suspension is infiltrated into leaves of *N. benthamiana*. A few days after infiltration leaves are harvested, and the protein of interest is purified. This figure was created with BioRender.com (accessed on 5 March 2022).

**Figure 2 plants-11-01093-f002:**
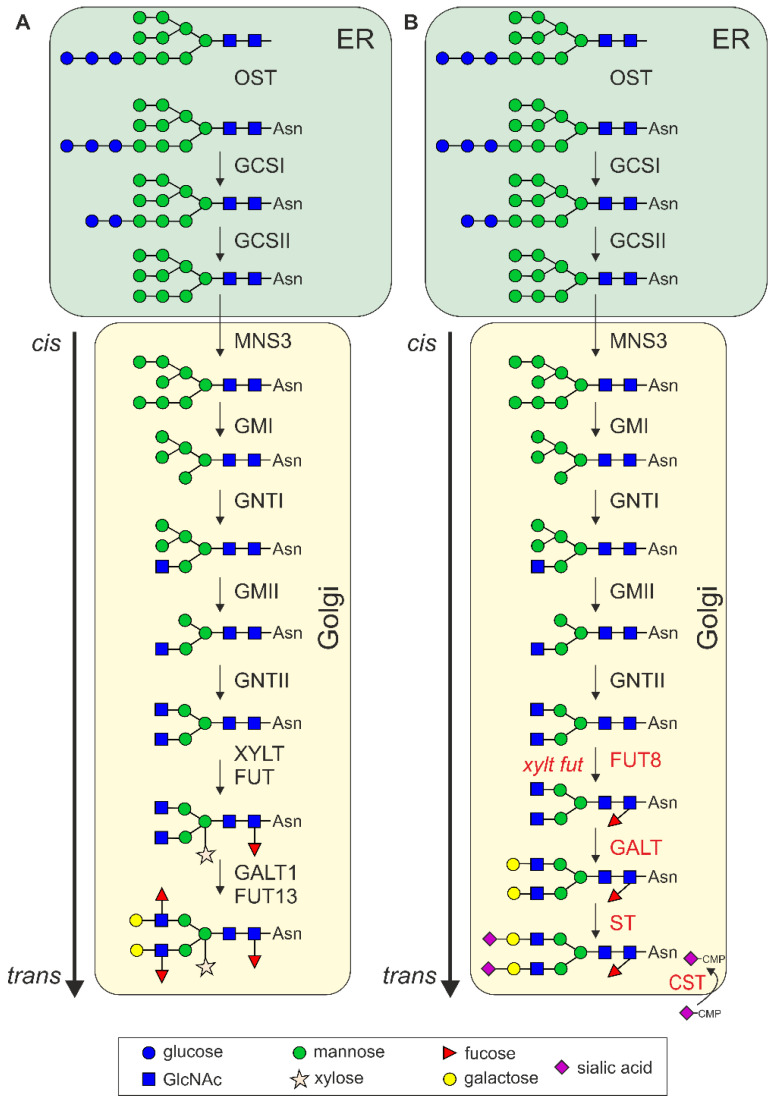
Schematic overview of natural and engineered *N*-glycan processing pathways. (**A**) *N*-glycan processing steps in the endoplasmic reticulum (ER) and Golgi apparatus of plants. OST, oligosaccharyltransferase complex; GCSI, α-glucosidases I (GCSI); GCSII, α-glucosidases II; MNS3, ER α-mannosidase; GMI, Golgi α-mannosidase I; GNTI, β1,2-*N*-acetylglucosaminyltransferase I; GMII, Golgi α-mannosidase II; GNTII, β1,2-*N*-acetylglucosaminyltransferase II; XYLT, β1,2-xylosyltransferase; FUT, core α1,3-fucosyltransferases; GALT1, Lewis type β1,3-galactosyltransferase; FUT13, α1,4-fucosyltransferase. (**B**) Engineered *N*-glycan processing pathway in plants. The knockout of XYLT and FUT (*xylt fut*) results in the formation of the GlcNAc_2_Man_3_GlcNAc_2_ (GnGn) structure, which serves as the acceptor substrate for β1,4-galactosyltransferase (GALT, β1,4-galactosylation) and core α1,6-fucosyltransferase (FUT8, core a1,6-fucosylation). Capping with terminal sialic acid is achieved by expression of α2,6-sialyltransferase (ST, a2,6-sialylation) together with the Golgi CMP-sialic acid transporter (CST) and proteins for CMP-sialic acid biosynthesis (not shown) [36,60].

**Figure 3 plants-11-01093-f003:**
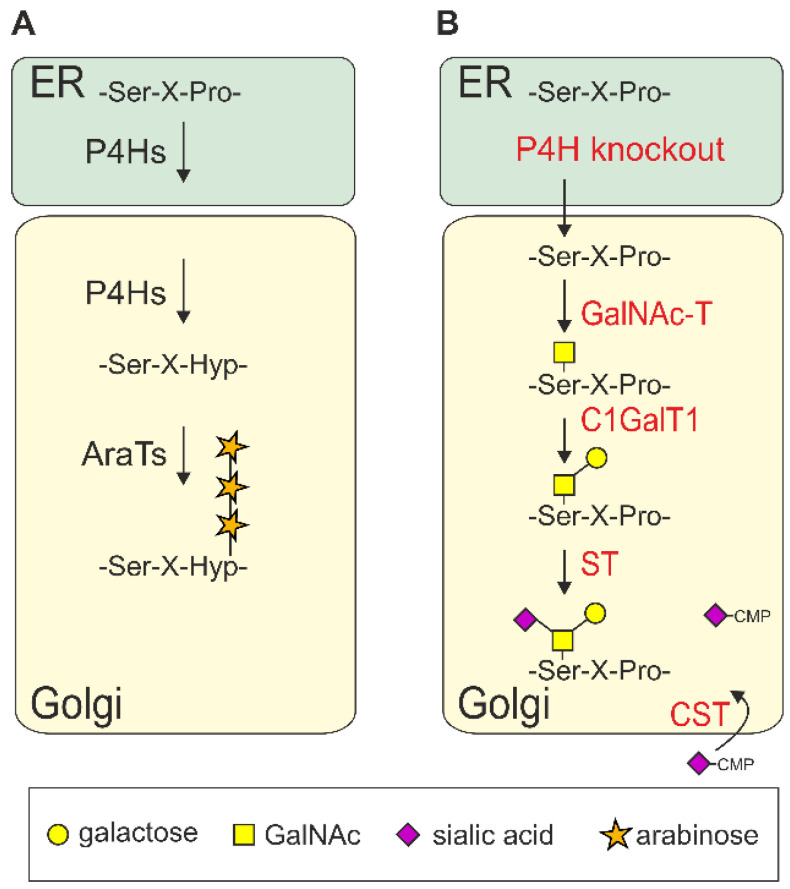
Schematic overview of natural and engineered *O-*glycosylation pathways. (**A**) Plant-type *O*-glycosylation. Proline residues next to Ser/Thr residues can be converted to hydroxyproline (Hyp) by prolyl-4-hydroxylases (P4Hs). Hyp residues are further elongated by arabinosyltransferases (AraTs). (**B**) Engineered mammalian-type *O*-glycosylation. The knockout of P4Hs prevents Hyp formation [67]. Expression of polypeptide GalNAc-transferases (GalNAc-T), β1,3-galactosyltransferase 1 (C1GalT1), sialyltransferase (ST), the Golgi CMP-sialic acid transporter (CST), and proteins for CMP-sialic acid biosynthesis (not shown) results in mucin-type *O*-glycan formation [37].

**Figure 4 plants-11-01093-f004:**
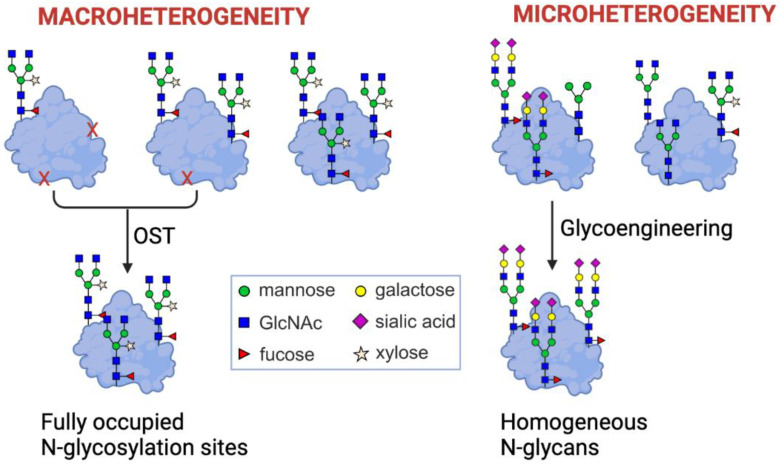
Both macroheterogeneity (variable occupancy of *N*-glycosylation sites) and microheterogeneity (variable *N*-glycan composition at distinct *N*-glycosylation sites) of glycosylation can have a negative impact on recombinant glycoprotein production and function. Macroheterogeneity can be overcome by engineering of the oligosaccharyltransferase (OST) complex. Microheterogeneity can be overcome by glycoengineering which involves elimination of unwanted glycan processing reactions and expression of glycosidases and glycosyltransferases for defined glycan structures. This figure was created with BioRender.com (accessed on 5 March 2022).

## Data Availability

Not applicable.

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
