# Peer review of "Transient Expression of Glycosylated SARS-CoV-2 Antigens in Nicotiana benthamiana"

_plants, 2022, doi:10.3390/plants11081093_

Round 1

Reviewer 1 Report

This manuscript gives an exhaustive description of the state of the art of transiently expressed glycosylated viral antigens especially related to SARS-CoV-2. I thinks it gives an  important contribution to the molecular farming field.  It is very well written and  easy to follow. I really have no major comments to make,

just one thing I would like to comment is that I would refer in the title and keywords specifically to SARS-CoV-2. In fact nearly all bibliographic references relate to this specific pathogen.

Author Response

Reply: Thank you for this comment, the title and keywords have been changed accordingly.

Reviewer 2 Report

The manuscript entitled "Transient expression of glycosylated viral antigens in Nicotiana benthamiana" describes recent developments in the transient production of viral glycoproteins in N. benthamiana, with particular emphasis on viral antigens derived from SARS-CoV-2. The entire manuscript is structured very clearly and presents all the issues covered in a way that is interesting to the reader.
In my opinion, the only point worth considering is to emphasize even more strongly the advantages of plant expression platforms used to produce heterologous proteins.

Author Response

Reply: Thank you for this comment, the advantages of plants are now more emphasized.

Reviewer 3 Report

see attachment

Author Response

This is a concise and timely review about the transient expression of viral glycoproteins in N. benthamiana. Given the ongoing Covid-19 pandemic, it is entirely reasonable that the authors have chosen to focus on SARS-CoV-2. However, so strong is the focus on this virus that the authors might wish to consider changing the title so that “SARS-CoV-2” replaces “viral” given that other viruses are only mentioned in passing. Overall, this is a useful contribution to the literature.

Reply: Thank you for this comment, the title and keywords have been changed accordingly.

I have a few comments aimed at improving the current version.

  1. Sentence starting Line 39: “human and animal viruses”. Humans ARE animals so change to either just animal or human and veterinary. Also the statement “Many enveloped….viruses” implies there are some which don’t have a glycoprotein coat. Is this actually true? Finally, mention should be made of the fact that the glycoproteins are embedded in a lipid bilayer of host origin to form the envelope.

Reply: Thank you, the text has been changed accordingly.

  1. Line 62: It would be more accurate to say Agrobacteria are transfected with plasmids carrying DNA for the target protein. As written, the sentence implies the bacteria naturally encode the viral antigens.

Reply: This has been changed.

  1. Line 79: I think it is important to define ΔXT/ΔFT at this point.

Reply: This has been changed.

  1. Section 4, starting line 144: This section would benefit hugely from a diagram. At present it is very difficult, if not impossible, to follow the pathway described.

Reply: Thank you for this comment, the N- and O-glycosylation pathways are now included as additional Figures 2 and 3.

  1. Section 5, starting line 225: Though most research has focussed on the S protein, the authors should mention that the envelope of SARS-CoV-2 contains 2 other proteins, Envelope (E) and Membrane (M) both of which are glycosylated and transient expression of which has been attempted in plants with the aim of producing virus-like particles. In this regard, the authors should consider adding two additional references, which were probably published after the current MS was submitted: Moon et al: Scientific Reports | (2022) 12:1005 | https://doi.org/10.1038/s41598-022-04883-y. Jung et al. Plant Biotechnol J. 2022 Mar 24. doi: 10.1111/pbi.13813

Reply: This is now briefly mentioned and the references have been added.

I also have some minor typographic corrections

Line 27: spurred on by more…

Line 36: contributes to solving

Line 46: tobacco-related

Line 86: grew normally..

Reply: These typographic errors have been changed accordingly.

Reviewer 4 Report

The manuscript is devoted to an extremely relevant topic - transient expression of viral proteins in plant cells and their glycosylation. The main attention is paid to the expression of SARS-CoV-2 antigens, the progress made in this area, the existing problems and ways to solve them. The work is clearly written, logical and easy understandable, the material may be of interest to a wide range of specialists. However, some minor changes are required.

Remarks:

There are not enough references in the introduction. It may be worth adding references to some statements, for example, to lines 41-44.

Section 4, dedicated to the description of glycosylation pathways, would probably benefit from a graphical presentation of information. Despite the fact that these processes are described in many reviews, a schematic representation of the glycosylation pathways would facilitate the understanding of the described processes and their differences in plant and mammalian cells.

Section 6 is more devoted to describing the processing and organization of N-glycans than to the actual strategies for improving the expression of glycoproteins. It is necessary either to add more information about the ways and means of glycoengineering, or to correct the title of the section.

Author Response

Reply: Thank you, additional references have been added to the introduction and the pathways are now included as additional Figures 2 and 3.